# Groundwater Quality Characterization in an Overallocated Semi-Arid Coastal Area Using an Integrated Approach: Case of the Essaouira Basin, Morocco

**Mohamed Ouarani [1,2,\*], Mohammed Bahir [1,3], David J. Mulla [2], Driss Ouazar [1], Abdelghani Chehbouni [1,4], Driss Dhiba [1], Salah Ouhamdouch [3] and Otman El Mountassir [3]**

1   International Water Research Institute (IWRI), Mohammed VI Polytechnic University (UM6P), Hay My Rachid, Ben Guerir 43150, Morocco; bahir@uca.ac.ma (M.B.); ouazard@gmail.com (D.O.); Abdelghani.Chehbouni@um6p.ma (A.C.); Driss.DHIBA@um6p.ma (D.D.)
2   Department of Soil, Water, and Climate, University of Minnesota, Upper Buford Circle, St. Paul, MN 55108, USA; mulla003@umn.edu
3   High Energy and Astrophysics Laboratory, Faculty of Sciences Semlalia, Cadi Ayyad University, Marrakech 40000, Morocco; salah.ouhamdouch@edu.uca.ma (S.O.); otman.elmountassir@ced.uca.ma (O.E.M.)
4   Institut de Recherche Pour le Développement (IRD), Unité Mixte de Recherche (UMR), Centre D'études Spatiales de la Biosphère (Cesbio), 31401 Toulouse, France
\*   Correspondence: Mohamed.ouarani@um6p.ma; Tel.: +212-677590821

**Abstract:** In this study, hydrogeochemical analyses were combined with geographic information system (GIS) tools to investigate salinization sources of groundwater in the downstream part of the Essaouira basin, and to analyze the spatiotemporal trends in groundwater quality. To assess groundwater suitability for drinking purposes, the quality of sampled water was compared with the World Health Organization (WHO) and the Moroccan guidelines. Wilcox and US salinity laboratory (USSL) diagrams were used to evaluate groundwater suitability for irrigation. Hydrogeochemical analyses revealed that groundwater is of Na-Cl and Ca-Mg-Cl types. The analyses of the correlation between the chemical elements showed that the water–rock interaction and the reverse ion exchange are the major processes impacting groundwater degradation in the study area. The study of groundwater suitability for drinking and irrigation purposes shows that groundwater quality in the study area is permissible, but not desirable for human consumption. Additionally, groundwater is permissible for agricultural use but with high-salinity hazards. The spatial distribution of the physicochemical elements shows a general upward gradient from the north to the south and from the east to the west. The trend in groundwater quality during the last five years shows a shifting in the quality from the mixed Ca-Mg-Cl to the Na-Cl type.

**Keywords:** coastal aquifer; semi-arid condition; groundwater salinization; reverse ion exchange; evaporites dissolution; hydrogeochemistry; Essaouira basin; Morocco

## 1. Introduction

Most of the world's available freshwater is stored underground (96% of freshwater, excluding frozen water in glaciers) [1]. Groundwater provides almost 50% of all drinking water worldwide and 43% of all consumptive agricultural use of water [1]. Unfortunately, this resource is under pressure, especially in arid and semi-arid regions where it faces many issues related to level depletion and quality degradation [2]. Groundwater problems become more challenging in coastal areas that are

generally areas of dense population and where the salinization by seawater intrusion represents an additional serious problem [3–5].

In Morocco, more than 40% of the irrigation water comes from groundwater and the area irrigated with groundwater contributes nearly 75% of the country's exports of high-value orchard and vegetable crops [6]. However, this resource is overallocated, especially in the coastal regions, as several studies have shown seawater intrusion within many Moroccan coastal aquifers [7–11].

Because of the surface water scarcity, groundwater is overallocated in the Essaouira basin. Based on data collected from the Tensift Hydraulic Basin Agency (ABHT)—the agency responsible for water resources management in the region—groundwater levels are sensitive to variation in precipitation and show a declining trend, as illustrated by Figure 1. The groundwater depletion in the study area is due not only to climate variability, but also due to anthropogenic impact, such as unmanaged pumping and the construction of a dam on the principal stream in the region. Those factors have increased the volume of water coming out of the aquifer and decreased its recharge. Groundwater is the main source for drinking water in the region. Tourism and agriculture are the major economic activities in the Essaouira region. Several touristic infrastructures and hotels are using groundwater. Based on data provided by the Essaouira Provincial Directorate of Agriculture (DPA), the agriculture sector in the Essaouira basin is dominated by small-scale irrigated areas, with farm areas varying between 5 and 80 hectares. However, the agricultural sector in the region is growing, especially in the framework of the "Moroccan Green Plan" and the "Green Generation Plan", by which the Moroccan government has been trying to develop the agriculture sector and better-value irrigation water.

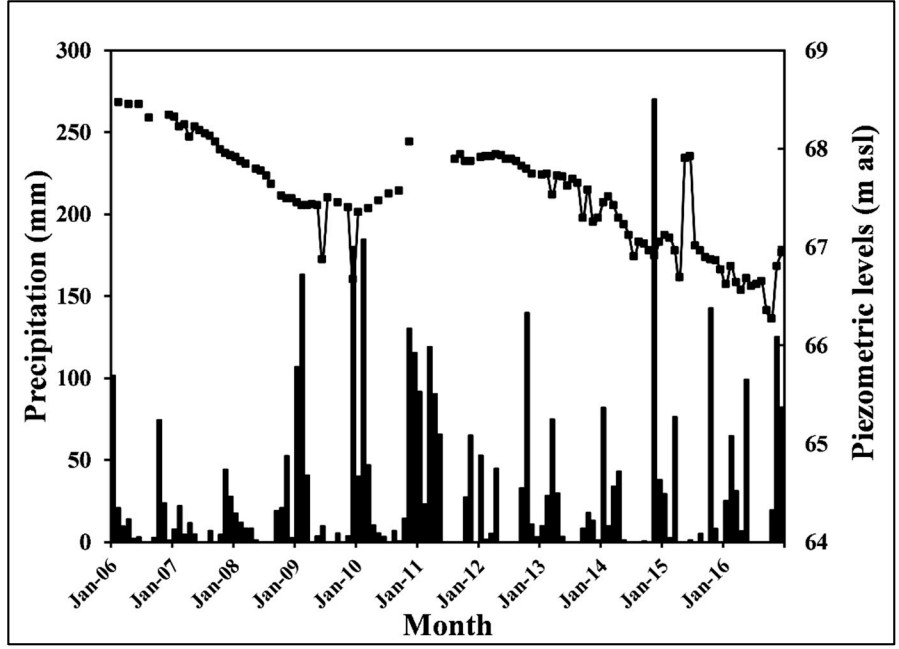

**Figure 1.** Groundwater level changes from 2006 to 2016.

This research aims: (1) to study the salinization sources of groundwater in the downstream part of the Essaouira basin using hydrogeochemical and GIS tools, (2) to assess the suitability of groundwater for drinking and irrigation uses in the study area, (3) to analyze the trend in water quality during the last five years based on data collected during 2015, 2017, 2018 and 2019 sampling campaigns carried out in the study area and (4) to make recommendations to preserve groundwater resources in the Essaouira basin.

This study would be of critical importance for the projects aiming to implement an integrated management plan for groundwater in the region of study. Additionally, this research will be useful for building a conceptual hydrogeological circulation model of groundwater in the region.

## 2. Materials and Methods

### 2.1. Study Area

The study area—known for "Essaouira bowl" formation—is in the downstream part of the Essaouira basin and covers an area of 350 km$^2$. It is bordered by the Ksob wadi (river) in the north, Tidzi wadi in the south, Tidzi diapir in the east and the Atlantic Ocean in the west (Figure 2a). The study area is characterized by low hills with a sparse hydrographic network [12]. The region has a semi-arid climate with a mean annual rainfall less than 300 mm/year and an annual mean temperature of about 20 °C [13]. The Plio-Quaternary aquifer, which is the upper unconfined aquifer, and the Turonian one, which is a confined aquifer, are the main groundwater resources in the region. The Plio-Quaternary is a sandstone-conglomerate aquifer with a variable thickness up to 60 m. The Turonian aquifer is a limestone aquifer of 60 m thickness. In most parts of the study area, the Turonian layer is confined by the impermeable Senonian layer whose thickness reaches 200 m in some locations, but the Plio-Quaternary and the Turonian aquifers are in direct contact by the edges of the synclinal formation (Figure 2b). This study will focus on the Plio-Quartenary aquifer, which is the shallow aquifer.

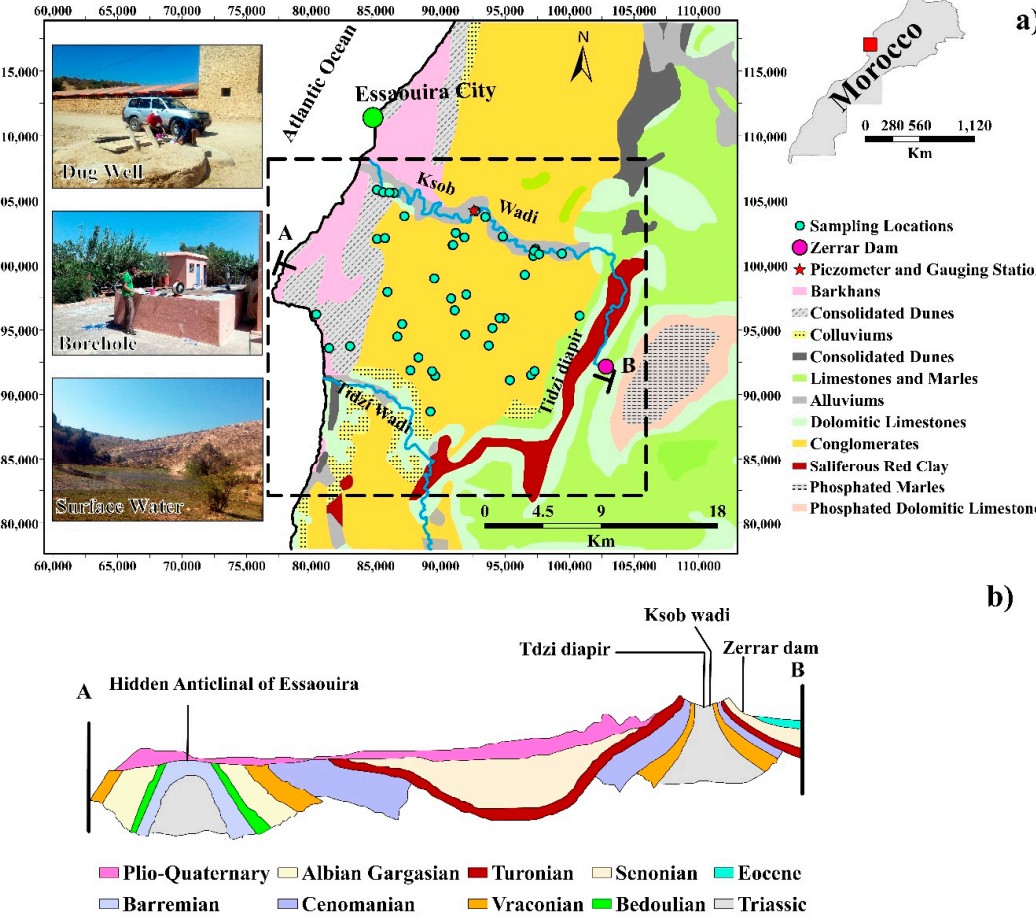

**Figure 2.** (**a**) Geographic location and geologic map (modified after References [12,14,15]). (**b**) Cross-section of the studied area (modified after References [12,16,17]).

### 2.2. Sampling and Analyses

113 samples were collected from surface water, wells and boreholes during a sequence of sampling campaigns conducted in 2015, 2017, 2018 and 2019 in the downstream part of the Essaouira basin. All the sampled wells are capturing the Plio-Quaternary aquifer. Whenever it is possible, the same wells/boreholes are sampled, however when a well sampled in the previous campaign goes dry

or becomes inaccessible, the closest well is sampled. Therefore, the collected samples were taken from 50 individual wells whose spatial distribution is presented in Figure 2a. The physico-chemical parameters (electrical conductivity (EC), pH, total dissolved solids (TDS) and temperature) were measured in the field using the multi-parameter meter HI9828. The depth of the water level was measured with a sound piezometric probe of 200 m.

Analyses of chemical elements (Ca, Mg, Na, K, Cl, $SO_4$ and $NO_3$) for the 2015 campaign were carried out in the Center of Analysis and Characterization in the Faculty of Sciences Semlalia of Marrakech using ion chromatography [11]. Alkalinity is determined by a pH meter using a sulfuric acid solution. For the campaign of 2017, analyses of chemical elements were conducted in the Laboratory of Geosciences and Environment at "Ecole Normale Superieure" (ENS) of Marrakech (Cadi Ayad University, Marrakech, Morocco) [17]. Cl concentrations were determined by Mohr titration. The concentrations of $HCO_3$ and $CO_3$ were determined with titration, using 0.1 N HCL. For the 2018 campaign, analyses of chemical elements, Na, K, Mg, Ca, Cl, $NO_3$ and $SO_4$, were performed using Inverse Liquid Chromatography (ILC) in the laboratory of Radio-Analysis and Environment (LRAE) at the National School of Engineers of Sfax, Tunisia [15]. The concentrations of $HCO_3$ and $CO_3$ were determined with titration, using 0.1 N HCL. For the 2019 campaign, the chemical analyses were conducted in Mohamed VI Polytechnic University of Benguerir-UM6P-(Morocco) labs. The samples analyses for Ca, Mg, Cl, $SO_4$, $NO_3$ and $HCO_3$ were performed using a SKALAR San++ Continuous Flow Analyzer (CFA), while Na and K concentrations were measured using Atomic Absorption Spectrophotometry (AAS).

The analyses results were validated based on the ionic balance. The sum of cations concentrations in milliequivalents per liter (meq/L) is equal to the sum of anions concentrations within an acceptable error range, ideally ±5% but up to ±10% is acceptable [18]. The formula below was used to compute the ionic balance (IB):

$$IB = 100 \times \frac{\sum \text{ meq Cations} - \sum \text{ meq Anions}}{\sum \text{ meq Cations} + \sum \text{ meq Anions}}$$

All the analyzed samples have an IB within the ±10 range. The results of the physico-chemical analyses are summarized in Table 1.

### 2.3. Hydrogeochemical Methods

Hydrogeochemical tools such as bivariate plots, Piper and Chadha diagrams have been widely used in different parts of the world, mainly to study groundwater composition and to understand geochemical processes controlling groundwater quality [16,19–29].

In this study, the Piper diagram was used to classify groundwater facies. The Piper diagram is a trilinear diagram to describe water chemistry developed firstly by Hill in 1940 [30] and modified by Piper in 1944 [31]. In the Piper diagram, major cations and anions are plotted in the two base ternary plots as milliequivalent percentages, before being projected onto the diamond field.

Bivariate plots of the major ions and Chadha diagram were leveraged to study the geochemical processes controlling groundwater quality in the study area. The Chadha diagram (modified Piper diagram) is a diagram developed by Chadha in 1999 to help in the geochemical classification of natural waters. In this diagram, the difference between alkaline earth and alkali metals in milliequivalent percentage (% meq) is plotted on the X axis and the difference between weak acidic anions and strong acidic anions in % meq is plotted on the Y axis [32].

To explore the impact of rock–water interaction on water composition in the studied region, saturation indices (SI) of halite, calcite, gypsum and dolomite minerals were calculated to examine the chemical equilibrium for the aforementioned minerals. "DIAGRAMMES" program version 6.61 was used to make the Piper diagram and to calculate saturation indices [33]. The DIAGRAMMES program

uses the PHREEQC code and its database phreeqc.dat to calculate saturation indices of minerals based on Equation (1) [34]:

$$SI = \log\left(\frac{IAP}{K_t}\right) \tag{1}$$

where IAP is ion activity product and $K_t$ stands for solubility constant. A negative SI suggests that groundwater is undersaturated and mineral dissolution is an ongoing process, while a positive SI indicates groundwater oversaturation and precipitation is the ongoing process. The calculated saturation indices are provided in Table S1 in the Supplementary Materials.

### 2.4. Geospatial Distribution of the Physicochemical Parameters

ArcGIS Pro, version 2.3.0 and its spatial analyst tool were leveraged to map samples' locations and to study the spatial distribution of groundwater quality parameters. The ordinary kriging algorithm was used to map the spatial distribution of the analyzed hydrochemical parameters. Kriging has been widely and usefully used in the hydrogeochemistry field for variable applications such as the assessment of the concentration of heavy metals in groundwater, the estimation of spatio-temporal variability of nitrate concentration in groundwater and many other applications [35–41]. Ordinary kriging, the most widely used kriging method, permits to estimate a value at a point of a region with a known variogram, using data close to the estimation location [42]. The spherical model and default proprieties of semi variogram set by ArcGIS were used: the lag size was set to default output cell size while the major range, the partial sill and the nugget values were calculated by ArcGIS.

### 2.5. Groundwater Suitability Assessment

To assess groundwater suitability for either drinking or irrigation uses, several different criteria such as TDS, % Na, EC and sodium adsorption ratio (SAR) were leveraged as appropriate.

Groundwater suitability for both drinking water and irrigation was assessed based on TDS values (Table 2). The value of TDS is a very good indicator of the degree of water mineralization [43]. According to David and DeWiest [44], when water has a TDS value below 500 mg/L, it is classified as a good quality water and it is desirable for drinking use, whereas when the TDS concentration is between 500 and 1000 mg/L, it could be used as a source for drinking water. When TDS is between 1000 and 3000 mg/L, water could be used for irrigation, however when TDS overcomes 3000 mg/L, water becomes unfit for drinking and irrigation uses.

Groundwater suitability for drinking water purposes was evaluated through comparison of the most important water quality parameters, such as pH, Na, Cl, K, $SO_4$ and $NO_3$ concentrations, with the World Health Organization (WHO) guidelines [45] and the Moroccan standards for drinking water (Table 3).

The percentage of sodium (% Na) is a widely used parameter to assess water suitability for irrigation uses [18,20]. The assessment of % Na is important because irrigating with water of high % Na could impair the soil structure and reduce its permeability [18,20]. Table 4 summarizes water classification based on % Na as proposed by Wilcox in 1955 [46]. The % Na was calculated based on the Equation (2) [47]:

$$\%Na = \frac{Na + K}{[Ca + Mg + Na + K]} \times 100 \tag{2}$$

The Sodium Adsorption Ratio (SAR) is another commonly used parameter to characterize sodium hazards which, at high levels, could cause long-term damage to soil structure and impede water absorption by crops [18,47]. Water classification based on SAR as presented by Richards in 1954 [48] is given in Table 5.

Wilcox's diagram [46] and the US salinity laboratory (USSL) diagram [48] were leveraged to get a deeper insight about groundwater suitability for irrigation in the study area. Wilcox and USSL diagrams have been widely applied in different parts of the world for the evaluation of groundwater suitability for irrigation [49–56].

The SAR was calculated based on the Equation (3) [46,48]:

$$SAR = \frac{Na}{\left(\frac{Ca + Mg}{2}\right)^{0.5}} \tag{3}$$

Parameters calculation and diagrams plotting were done using Excel Microsoft office Proplus-en-us version.

## 3. Results and Discussion

### 3.1. The Physicochemical Parameters

Table 1 summarizes the descriptive statistics of the physico-chemical analyses by year. The fully detailed results of the physico-chemical analyses are given in Table S2 in the Supplementary Materials. Temperatures vary between 20.6 and 27.5 °C, between 17.6 and 26.3 °C, between 18.6 and 27.4 °C, and between 20.4 and 27.1 °C for 2015, 2017, 2018 and 2019 campaigns, consecutively. The temperature of groundwater samples is similar to the air temperature in the study area, due to the shallowness of the sampled aquifer. The collected samples during the four sampling campaigns show basic pH values. pH values vary between 7.5 and 8.2 for the 2015 campaign, between 7.1 and 9.0 for the 2017 campaign, between 7.2 and 8.7 for the 2018 campaign and between 7.3 and 8.9 for the 2019 campaign. The measured pH could be explained by the fact that 97% of the analyzed samples have the Ca-Mg-Cl or the Na-Cl chemical facies (Piper Diagrams; Figure 3).

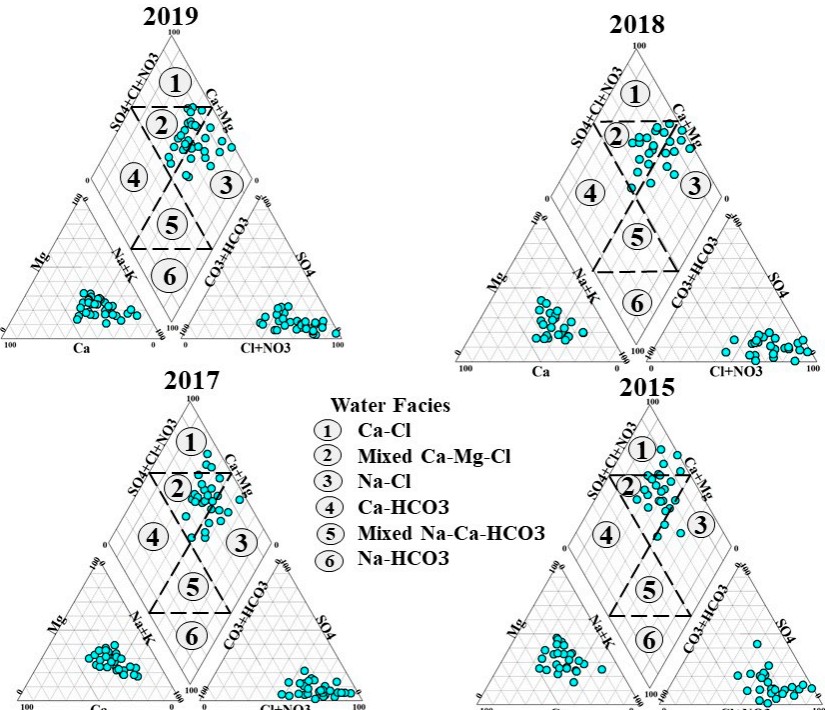

**Figure 3.** Piper diagrams showing hydrogeochemical facies of Plio-Quaternary aquifer within the Essaouira basin for the four sampling campaigns.

**Table 1.** Range, average and median of the physicochemical parameters of the analyzed samples from the Plio-Quaternary Aquifer (2015, 2017, 2018 and 2019).

| | pH | T (°C) | EC (µs/cm) | Ca (mg/L) | Mg (mg/L) | Na (mg/L) | K (mg/L) | HCO$_3$ (mg/L) | Cl (mg/L) | SO$_4$ (mg/L) | NO$_3$ (mg/L) |
|---|---|---|---|---|---|---|---|---|---|---|---|
| | | | | | | 2019 Campaign | | | | | |
| Range | 7.6–9.0 | 20.4–27.1 | 863–16,300 | 47.2–355.7 | 16.7–402.4 | 88.6–3546.5 | 1.7–118.9 | 167.8–404.5 | 113.0–5830.0 | 44.1–388.0 | 0.5–200.2 |
| Average | 8.2 | 23.0 | 2760 | 144.2 | 74.2 | 387.3 | 12.0 | 289.8 | 728.9 | 124.3 | 58.6 |
| Median | 8.4 | 22.7 | 2011 | 129.7 | 63.5 | 239.9 | 5.7 | 277.6 | 415.3 | 110.7 | 36.8 |
| | | | | | | 2018 Campaign [15] | | | | | |
| Range | 7.2–8.7 | 17.6–26.3 | 916–9744 | 58.1–364.7 | 14.6–238.2 | 75.9–1464.5 | 0–66.5 | 12.2–549.0 | 99.3–3158.9 | 4.8–408.3 | 0.0–396.8 |
| Average | 7.9 | 22.1 | 2816 | 140.5 | 71.7 | 293.0 | 12.2 | 263.5 | 680.7 | 122.4 | 41.4 |
| Median | 7.9 | 22.0 | 2176 | 100.2 | 70.5 | 181.6 | 7.8 | 262.3 | 439.6 | 120.1 | 18.6 |
| | | | | | | 2017 Campaign [17] | | | | | |
| Range | 7.1–9.0 | 18.9–26.4 | 724–7555 | 44.9–368.7 | 31.1–205.1 | 85.0–1430.9 | 4.4–74.5 | 115.9–567.4 | 170.4–3138.2 | 11.3–257.2 | 3.2–126.5 |
| Average | 7.7 | 22.1 | 2175 | 131.0 | 77.6 | 258.4 | 14.4 | 323.1 | 665.5 | 93.5 | 33.3 |
| Median | 7.6 | 22.2 | 1935 | 113.8 | 69.0 | 185.1 | 10.0 | 317.3 | 511.2 | 90.0 | 20.8 |
| | | | | | | 2015 Campaign [11] | | | | | |
| Range | 7.5–8.2 | 20.6–27.5 | 626–7840 | 44.9–391.2 | 17.9–191.1 | 27.3–461.0 | 11.0–226.8 | 11.3–563.7 | 45.9–1912.7 | 6.1–466.7 | 0.0–237.4 |
| Average | 7.7 | 22.1 | 2509 | 131.8 | 81.1 | 134.8 | 96.3 | 280.5 | 479.6 | 130.5 | 57.0 |
| Median | 7.8 | 22.3 | 2235 | 110.9 | 84.7 | 114.6 | 101.1 | 282.1 | 352.7 | 125.0 | 25.8 |

The electrical conductivity (EC) ranges between 626 and 7555 µS/cm for the 2015 campaign, between 724 and 7555 µS/cm for the 2017 campaign, between 916 and 9744 µS/cm for the 2018 campaign and between 863 and 16,300 µS/cm for the 2019 campaign. The wide range of physicochemical parameters' variability in the study area indicates variable levels of water salinization/mineralization that could result from different mechanisms, such as water–rock interaction, reverse ion exchange, seawater intrusion or anthropogenic source. Further explanation regarding the spatial distribution of those parameters and underlying geochemical processes is provided in Section 3.5, entitled "Spatial Distribution of Groundwater Quality Parameters". The different hydrogeochemical tools were leveraged to investigate the processes behind the degradation of groundwater quality in the study area.

The results of the geochemical analysis for the groundwater samples show the following order of cation dominance: $Na^+ > Ca^{2+} > Mg^{2+} > K^+$ and $Cl^- > HCO_3^- > SO_2^{4-} > NO_3^-$, for anions, and the dominance order remains the same for all the four campaigns. The average values of the physico-chemical parameters are used later in this report to study the temporal trend of water quality during the last five years.

### 3.2. Hydrogeochemical Facies

Piper diagrams in Figure 3 illustrate the chemical facies of the collected samples during the four sampling campaigns. All the samples collected in 2018 and 2019 are of Na-Cl and mixed Ca-Mg-Cl types (60% of the collected samples are of Ca-Mg-Cl types and 40% are of Na-Cl types for both campaigns). For the samples collected in 2015 and 2017, they are of Ca-Cl, Na-Cl and Ca-Mg-Cl types, with a predominance of Ca-Mg-Cl type, as illustrated by Figures 3 and 4. The Na-Cl type could be attributed to seawater intrusion, while the Ca-Cl and Ca-Mg-Cl types could be explained by ion exchange mechanisms between groundwater and subsurface geologic strata (e.g., limestone). The Chadha diagram (Figure 4) was used to understand the major geochemical processes behind the groundwater chemistry in the study area. The Chadha diagram confirms the predominance of the Ca-Mg-Cl type for the four sampling campaigns (85%, 67%, 60% and 58% of 2015, 2017, 2018 campaign samples, respectively).

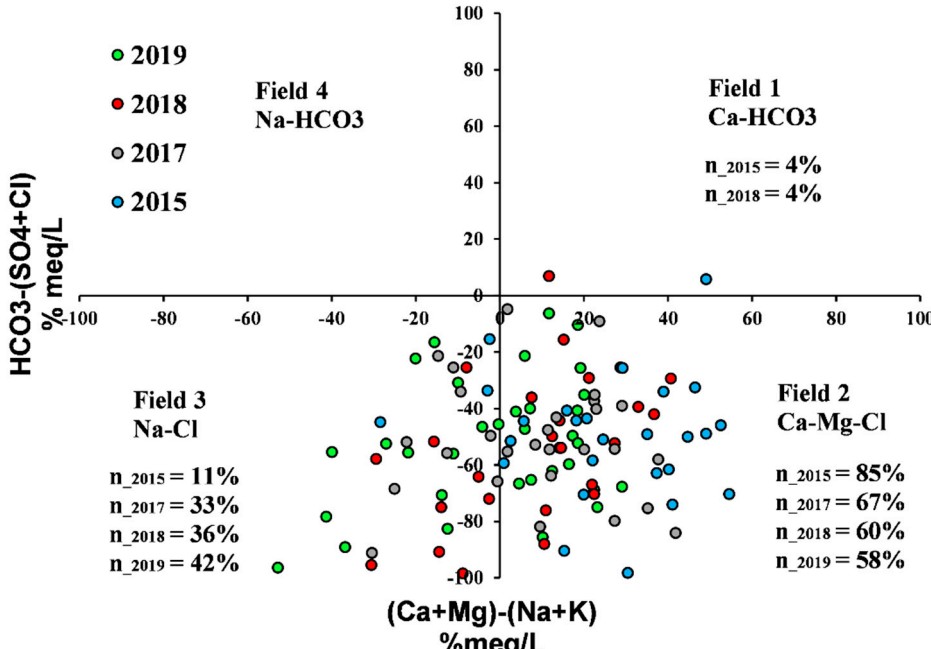

**Figure 4.** Chadha Diagram for the Groundwater Classification in the Essaouira Basin (2015, 2017, 2018, 2019).

### 3.3. The Hydrogeochemical Processes Controlling Groundwater Quality

To study more deeply the processes behind the geochemical features of the groundwater samples, hydrochemical relationships between different parameters were plotted (Figures 5 and 6). The 1:1 straight line in Figure 5 plots represents evaporite minerals dissolution for halite, calcite, gypsum and dolomite. There is a strong correlation between Na and Cl, $R^2 = 0.84$, $R^2 = 0.85$, $R^2 = 0.97$ and $R^2 = 0.97$, for the sampling campaigns of 2015, 2017, 2018 and 2019 respectively (Figure 5a), which means that the two minerals have the same origin, and this hypothesis is corroborated by the negative values of halite SI and the parabolic relationship between Na + Cl and the halite saturation indices (SI) (Figure 6b), suggesting halite dissolution as a major process in the study area. Halite dissolution is not the only source of Na and Cl, since many samples do not fall on the equiline. All those samples (not falling in the equiline) show a dominance of Cl compared to Na, which could be attributed to Na depletion due to the reverse ion exchange process. Minor correlation was found between $HCO_3^-$ and ($Ca^{2+} + Mg^{2+}$) and between $HCO_3^-$ and Ca respectively (Figure 5c,d), indicating that those ions came from a source other than dolomite and calcite dissolution. The high concentrations of Ca compared with $HCO_3^-$ could be explained by the reverse exchange that goes with seawater intrusion. The correlation diagram of Ca and $SO_4^{2-}$ (Figure 5b) shows that two types of groundwater could be categorized: a group of samples falling around and on the 1:1 line, suggesting the contribution of gypsum dissolution in groundwater mineralization. This hypothesis is strengthened by the parabolic proportional relationship in the plots of $Ca^{2+} + SO_4^{2-}$ vs. SI of gypsum and the negative gypsum SI (Figure 6b). Nevertheless, the parabolic pattern is less obvious compared with halite SI. The second group is represented by points located above the equiline with $Ca^{2+}$ predominance, which points to reverse ion exchange.

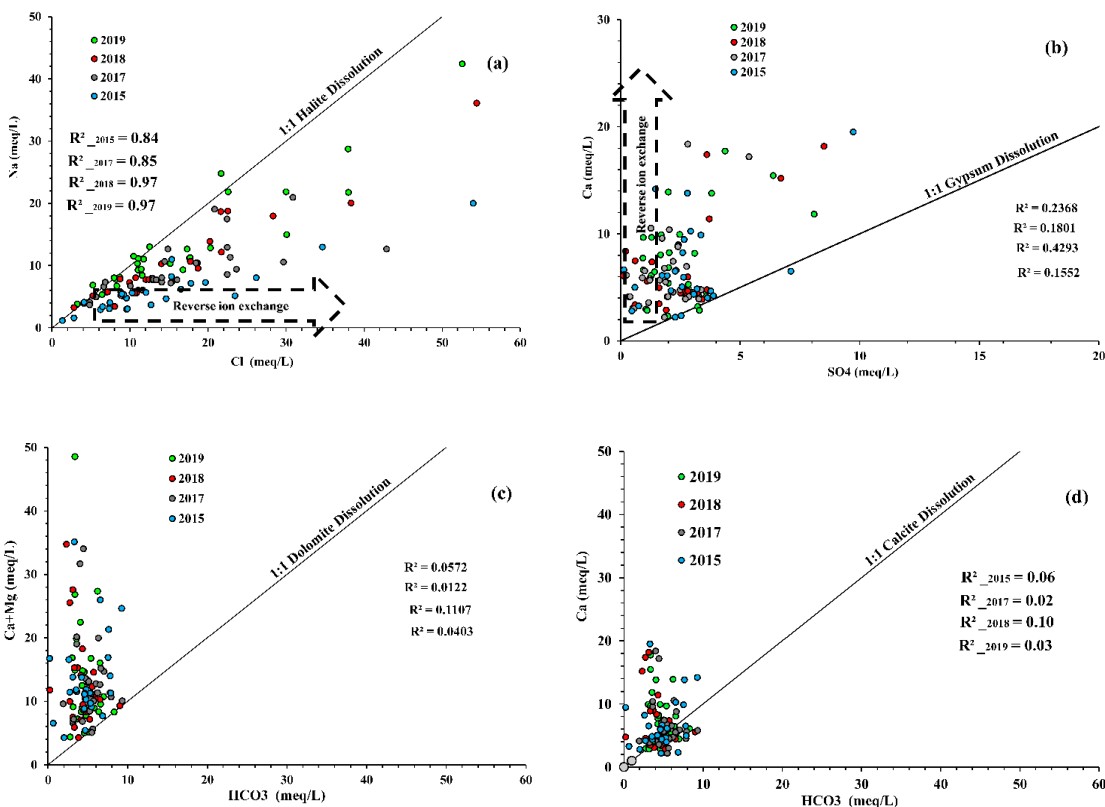

**Figure 5.** Correlation Diagrams: (**a**) Na vs. Cl, (**b**) Ca vs. $SO_4$, (**c**) (Ca + Mg) vs. $HCO_3$, (**d**) Ca vs. $HCO_3$.

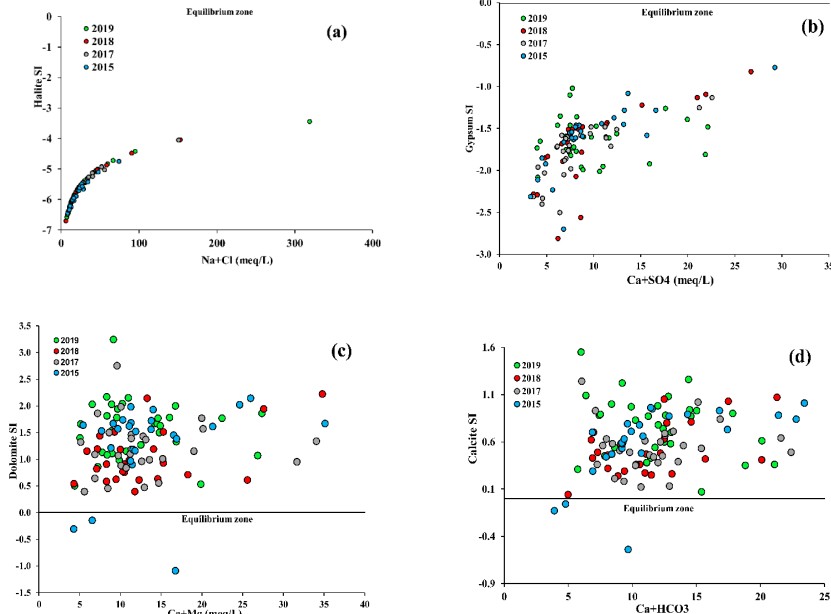

**Figure 6.** Relationships between: (**a**) (Na + Cl) and Halite SI, (**b**) (Ca + SO4) and Gypsum SI, (**c**) (Ca + Mg) and Dolomite SI, (**d**) (Ca + HCO3) and Calcite SI. (SI = Saturation indices).

### 3.4. Groundwater Suitability

Groundwater quality was classified based on TDS (Table 2). For the four campaigns, the majority of samples have TDS between 500 and 3000 mg/L, which is permissible for drinking uses and acceptable for irrigation. Only few points have either a TDS below 500 mg/L, which is of a good quality for drinking uses, or greater than 3000 mg/L, which unsuitable for both drinking and irrigation. Only one sample from the 2015 campaign, two points in 2017 and 2018 and one point in 2019 that have TDS exceeding 3000 mg/L and all those sampling locations except one are very close to the ocean, which suggests the impact of seawater intrusion. Based on TDS classification, water in the study area is more suitable for irrigation rather than for drinking uses.

**Table 2.** Groundwater classification based on TDS values [44]. (TDS = Total Dissolved Solids).

| TDS (mg/L) | Water Class | Percentage of Samples | | | |
|---|---|---|---|---|---|
| | | **2015** | **2017** | **2018** | **2019** |
| <500 | Desirable for drinking | 8 | 10 | 4 | 6 |
| 500–1000 | Permissible for drinking | 27 | 38 | 48 | 42 |
| <3000 | Acceptable for irrigation | 61 | 48 | 40 | 49 |
| >3000 | Unfit for drinking and irrigation | 4 | 4 | 8 | 3 |

Groundwater quality of the sampled wells/boreholes was compared with WHO and Moroccan guidelines and the results are given in Table 3. Most of the collected samples have a pH falling in the range of guidelines, and less than 4% of the samples are unsuitable for human consumption based on guidelines related to the pH. Between 81% and 92% of collected samples exceed the WHO guidelines for Cl, whereas between 15% and 31% of samples exceed the Moroccan standards. For sodium and potassium between 12% and 64%, and 21% and 92% of collected samples respectively, are unfit for drinking use based on WHO guidelines, 6% to 8% of samples have $SO_4$ exceeding the WHO guidelines and less than 4% of samples exceed the Moroccan standards. While 20% to 45% of samples exceed the WHO and the Moroccan guidelines for $NO_3$, 14% to 16% of samples are unfit for drinking uses based on WHO guidelines for Ca. This comparison with WHO and the Moroccan standards reveals the mediocre quality of groundwater with respect to drinking purposes.

**Table 3.** Water parameters comparison with World Health Organization (WHO) and Moroccan guidelines.

| Parameter | WHO 2011 Guidelines | Moroccan Standards | Percentage of Samples Exceeding the Limit of WHO Guidelines (%) | | | | Percentage of Samples Exceeding the Moroccan Standards Limit (%) | | | |
|---|---|---|---|---|---|---|---|---|---|---|
| | | | 2015 | 2017 | 2018 | 2019 | 2015 | 2017 | 2018 | 2019 |
| pH | 6.5–8.5 | 6.5–8.5 | 0 | 3 | 4 | 0 | 0 | 3 | 4 | 0 |
| Cl (mg/L) | ≤250 | ≤750 | 81 | 83 | 92 | 82 | 15 | 31 | 28 | 24 |
| Na (mg/L) | ≤200 | – | 12 | 41 | 44 | 64 | – | – | – | – |
| K (mg/L) | ≤12 | – | 92 | 34 | 24 | 21 | – | – | – | – |
| $SO_4$ (mg/L) | ≤250 | ≤400 | 8 | 3 | 8 | 6 | 4 | 0 | 4 | 0 |
| $NO_3$ (mg/L) | ≤50 | ≤50 | 42 | 24 | 20 | 45 | 42 | 24 | 20 | 45 |
| Ca (mg/L) | ≤200 | – | 15 | 14 | 16 | 15 | – | – | – | – |

Irrigation water quality classification based on % Na and on SAR were used to examine the suitability of groundwater for irrigation, and the results summary is given in Tables 4 and 5. Based on % Na classification, the majority of the analyzed samples have good to permissible quality in regard to irrigation uses (20% < % Na < 60%) but none of the samples fall in the excellent class (% Na < 20%) or in the unsuitable class. Regarding the hazards related to SAR, they are very low since between 88% to 100% of samples have excellent quality, less than 9% have good quality and less than 3% of the samples have permissible quality, and none of the samples fall in the doubtful class based on SAR classification. Based on the Wilcox diagram (Figure 7), 47% of the collected samples fall in the "good to permissible" and "the permissible to doubtful" domains, 27% of samples fall in the "doubtful to unsuitable" field and 22% of points are unsuitable for irrigation uses.

**Table 4.** Groundwater quality classification based on % Na [46].

| Water Quality | Percentage of Sodium (%) | Percentage of Samples (%) | | | |
|---|---|---|---|---|---|
| | | 2015 | 2017 | 2018 | 2019 |
| Excellent | <20 | 0 | 0 | 0 | 0 |
| Good | 20–40 | 61 | 42 | 28 | 12 |
| Permissible | 40–60 | 35 | 48 | 64 | 67 |
| Doubtful | 60–80 | 4 | 10 | 8 | 21 |
| Unsuitable | >80 | 0 | 0 | 0 | 0 |

**Table 5.** Groundwater quality classification based on SAR [47,48]. (SAR = Sodium Adsorption Ratio).

| Water Quality | SAR Values | Percentage of Samples (%) | | | |
|---|---|---|---|---|---|
| | | 2015 | 2017 | 2018 | 2019 |
| Excellent | <10 | 100 | 97 | 96 | 88 |
| Good | 10–18 | 0 | 3 | 4 | 9 |
| Permissible | 18–26 | 0 | 0 | 0 | 3 |
| Doubtful | >26 | 0 | 0 | 0 | 0 |

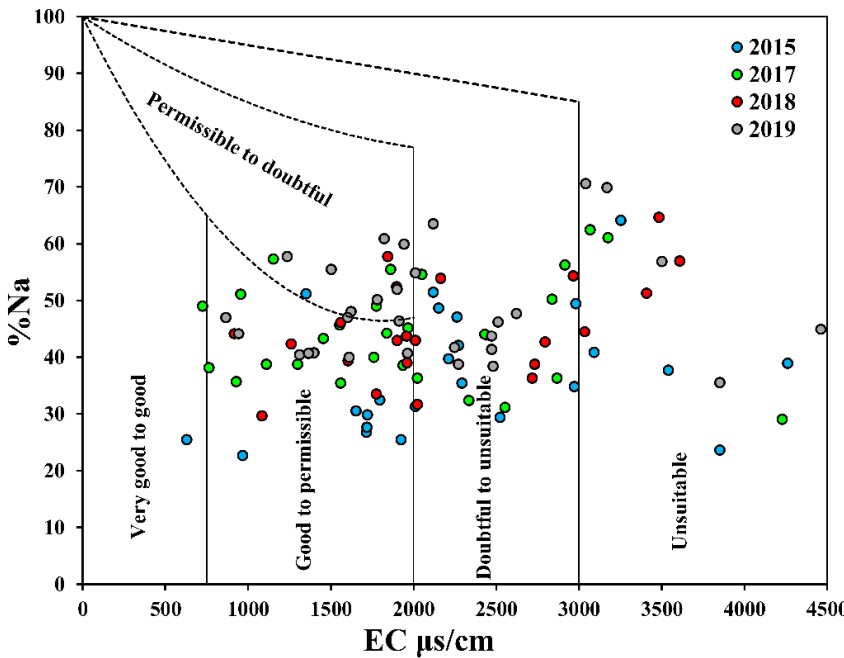

**Figure 7.** Classification of irrigation water quality based on EC and % Na.

The USSL diagram in Figure 8 shows that 49% of the total number of samples collected during the four sampling campaigns fall into the C3S1 class corresponding to low SAR hazard and high salinity

risk. Water falling in the C3 class should be only used on soils with good leaching abilities and with plants of good salinity tolerance, and requires special management to control salinity [57].The study of water suitability for irrigation shows low hazards related to SAR and % Na for almost the totality of sampled wells, however around 89% of samples show high to very high salinity hazards.

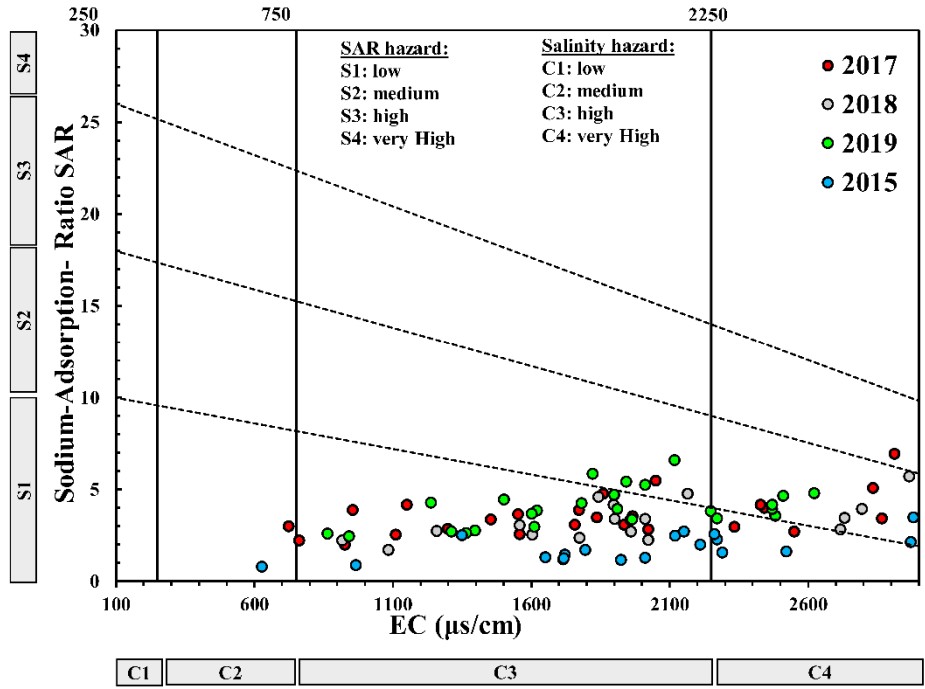

**Figure 8.** Classification of irrigation water quality based on EC and SAR.

### 3.5. Spatial Distribution of Groundwater Quality Parameters

Data from the newest campaign (2019) was used to map the spatial distribution of the physicochemical parameters (Cl, Na, K, Ca, $SO_4$, $NO_3$, EC and TDS) in the study area (Figure 9). Cl, Na, K, Ca, EC and TDS show an increasing gradient going from the North in the recharging area by the Ksob river toward the south, and from the east toward the west, where the ocean is located. The lowest values of Cl, Na, TDS and EC are observed in the east and north parts and in the south-west close the Tidzi river and increases toward the middle and the west parts of the study area. The maximal Cl, Na, TDS and EC values were recorded at well N°1, which is the closest one to the ocean. This could be explained by the increasing impact of seawater intrusion and reverse ion exchange when getting closer to the ocean. This variability is more explicit for EC values that are about ten times higher at the closest well compared to some other wells further inland. The lowest Ca values are observed along Ksob wadi and increase gradually toward the middle and the south-west of the study area, where the highest Ca concentrations are recorded. $SO_4$ and $NO_3$ show higher concentrations in specific isolated spots. The highest $SO_4$ concentrations are recorded in samples N°23 and N°1. For $NO_3$, the highest concentrations were observed at sampling locations 1, 20, 26, 27, 28 and 32. The high $SO_4$ concentrations at locations 23 and 1, besides the high $NO_3$ concentrations at the points 1, 20, 28 and 32, could be explained by the heavy touristic activity in those areas (especially Sidi Kaouki zone) combined with the absence of a wastewater treatment plant. The high $NO_3$ concentrations at the points 26 and 27 could be attributed to the livestock activities in the areas close to those points. Based on the interpretation of the maps in Figure 9, three factors are impacting the spatial distribution of groundwater parameters: the distance away from the ocean (Cl, Na, K, EC and TDS), the evaporite minerals dissolution (Cl, Na, K, EC and TDS) and the anthropogenic impact through groundwater pollution by sewage and livestock activities in the region ($SO_4$ and $NO_3$).

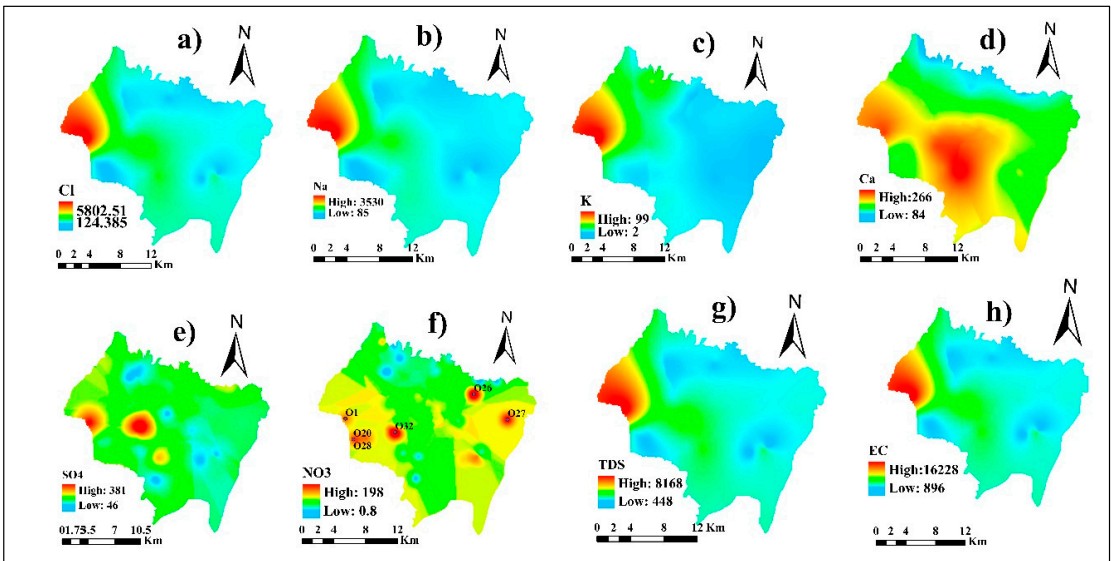

**Figure 9.** Spatial distribution of groundwater physicochemical parameters ((**a**) Cl, (**b**) Na, (**c**) K, (**d**) Ca, (**e**) SO$_4$, (**f**) NO$_3$, (**g**) TDS and (**h**) EC) in the Essaouira Basin based on the 2019 campaign.

### 3.6. Groundwater Quality Trend in the Study Area

The average of elements concentration computed based on data collected during the four sampling campaigns are plotted in Figure 10 Cl and Na concentrations exhibit a clear increasing tendency during the last five years. The average concentration of Cl increased from 479.6 mg/L in 2015 to 728.9 mg/L in 2019, while the mean concentration of Na increased from 134.8 mg/L in 2015 to 387.3 mg/L in 2019. This remark is corroborated by the shifting toward the Na-Cl type illustrated by both Piper and Chadha diagrams (Figures 3, 4 and 11). Figure 11 illustrates the distribution of hydrochemical facies in each of the sampling campaigns based on Piper diagram results. The percentage of Na-Cl increased from 10% in 2015 to 40% in 2018 and 2019, passing by 20% in 2017. The same tendency is shown by the Chadha diagram since the percentage of samples falling within the Na-Cl field has gradually increased from 11% in 2015 to 33% in 2017, to 36% in 2018 and to 42% in 2019. The shift in the groundwater chemistry toward the Na-Cl type could be attributed to the fact that more interaction is occurring between seawater and the groundwater in the study area, probably due to the aggravation of seawater intrusion. This shifting in the groundwater chemistry has impacted groundwater suitability to irrigation as the maximal values of % Na and SAR were observed within 2019 campaign records, while the minimal values were recorded during the 2015 sampling campaign. No clear temporal trend has been observed for the other chemical elements. Based on the parameters used to assess groundwater appropriateness for drinking and irrigation uses (Tables 2–5), we were not able to see the effect of changes in water quality on its suitability. This might be due to the short time period of the sampling (5 years). Therefore, a longer monitoring of groundwater quality in the study area is recommended to get a greater insight about the temporal trend of changes in groundwater quality and its suitability.

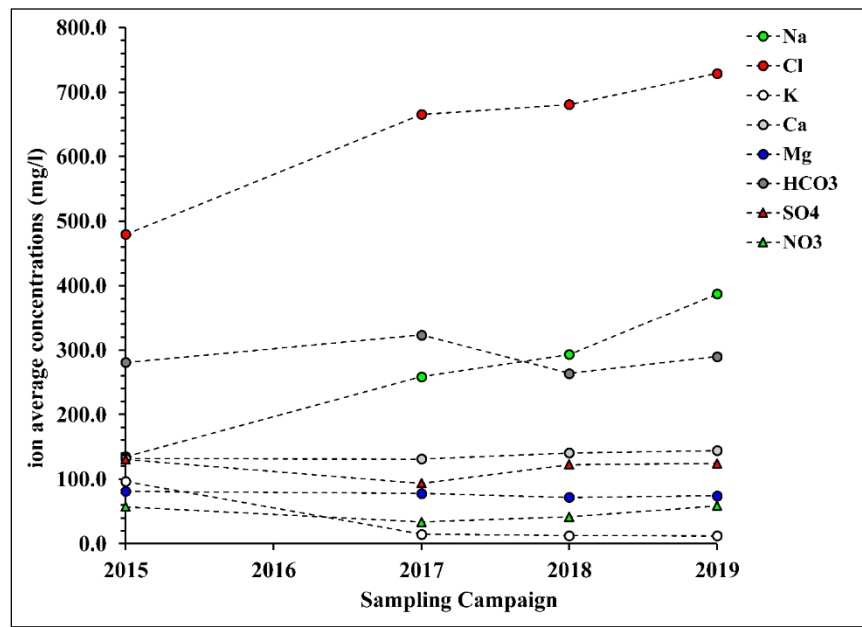

**Figure 10.** Variation of the chemical composition of groundwater in the Essaouira basin during the last five years.

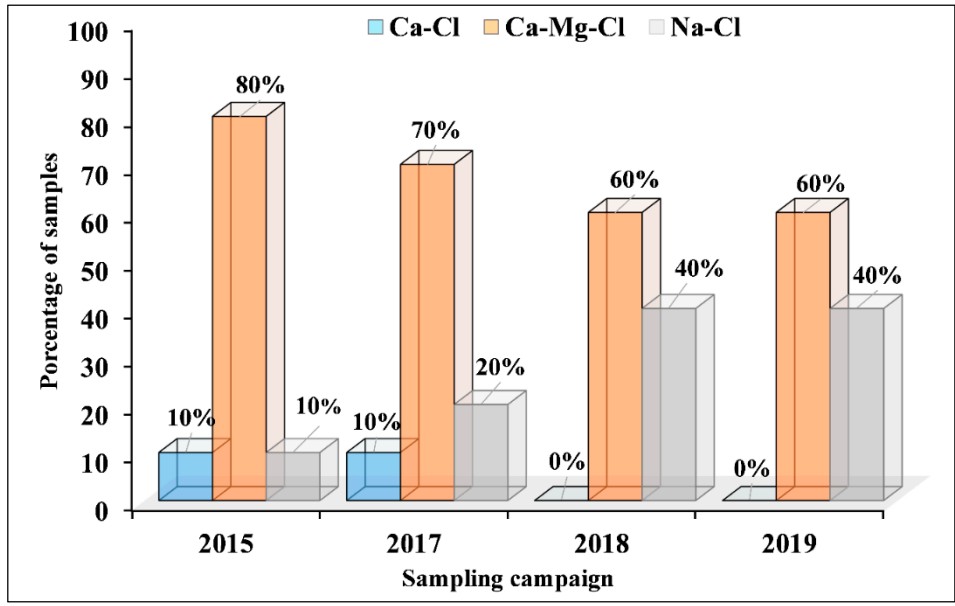

**Figure 11.** Distribution of hydrochemical facies for the four sampling campaigns.

## 4. Conclusions

This study has shown how useful the combined use of the hydrogeochemical approach and GIS tools is in exploring the mechanisms behind groundwater salinization, depicting groundwater spatiotemporal variation and evaluating its quality for drinking and irrigation uses. Hydrogeochemical analyses revealed that groundwater in the downstream part of the Essaouira basin is of Na-Cl and mixed Ca-Mg-Cl types. The bivariate plots representing correlation between the chemical elements showed that the water–rock interaction and the reverse ion exchange accompanying seawater intrusion are the major processes that impact groundwater degradation in the study area. The study of groundwater suitability for drinking and irrigation purposes shows that groundwater is permissible but not desirable for human consumption, mainly due to its high Cl concentration exceeding the WHO

and the Moroccan guidelines. On the other hand, groundwater is permissible for irrigation use but with high salinity hazards. Therefore, special management and selecting plants tolerant for salinity are needed when irrigating with groundwater from the studied aquifer. The spatial distribution of the physicochemical elements in the study area shows a general upward gradient from the north to the south and from the east to the west. The trend in groundwater quality during the last five years shows a shifting in the quality from the mixed Ca-Mg-Cl to the Na-Cl type, which could point to an aggravation in the seawater intrusion in the coastal area. The results of this study pointed to groundwater degradation in the study area during the last five years. Therefore, measurements to reduce pressure on this resource are required. We recommend taking into consideration groundwater recharge in the management of the dam by maintaining a regular flow in the river. Regulation of groundwater abstraction, especially from the closest wells to the ocean, is recommended to mitigate seawater intrusion in the coastal part of the aquifer. Additionally, we recommend conducting longer monitoring of groundwater quality in the region to get more insight about the temporal trend of changes in groundwater quality. Lastly, we recommend building a numeral model of this coastal aquifer to demark more accurately the seawater extent and to simulate the impact of variable scenarios of water development and climate change on water resources in the study area.

**Supplementary Materials:** The following are available online at http://www.mdpi.com/2073-4441/12/11/3202/s1, Table S1: Calculated Saturation Indices of the Analyzed Samples (2015, 2017, 2018, 2019), Table S2: Physicochemical Parameters of the Analyzed Samples from the Plio-Quaternary Aquifer (2015, 2017, 2018 and 2019).

**Author Contributions:** Conceptualization, M.O., M.B., D.J.M. and D.O.; methodology: M.O., M.B. and D.J.M.; software, M.O.; validation, M.B. and D.J.M.; formal analysis and investigation, M.O., S.O. and O.E.M.; resources: D.O. and A.C.; writing—original draft preparation: M.O.; writing—review and editing: D.J.M., M.B. and M.O.; visualization: M.O.; supervision: M.B., D.J.M., A.C. and D.D.; project administration: A.C. and D.D. All authors have read and agreed to the published version of the manuscript.

**Funding:** This research received no external funding.

**Acknowledgments:** We deeply appreciate the useful remarks and comments of the reviewers. We would also like to thank people from the Essaouira Provincial Directorate of Agriculture (DPA) and the National Office of drinking water (ONEE) in the Essaouira region for their assistance. Namely, we thank Aziz Soulaimani, the Technical Manager of the Water, Soil and Agriculture Analysis Laboratory within the UM6P, for his assistance during the chemical analyses. We do not forget to thank Issam Elkessab, the head of the Water Management and Rural Infrastructure department at the Essaouira DPA for his time and guidance.

**Conflicts of Interest:** The authors declare no conflict of interest.

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
