# Peer review of "Groundwater Quality Characterization in an Overallocated Semi-Arid Coastal Area Using an Integrated Approach: Case of the Essaouira Basin, Morocco"

_water, doi:10.3390/w12113202_

Round 1

Reviewer 1 Report

The manuscript as received has some hidden information; overlapping of figures and miss-siting of table 1, in addition to some structural mistakes, all noted in the attached file. Since the missed information are crucial, the scientific quality of paper can't be fully judged at this review cycle, please resubmit a corrected version.

Author Response

Thank you for your remarks.

I see that the table 1 and some Figures failed when producing the pdf file, however the word file was correct. I fixed this issue, besides the formatting problems related to em-dash symbols in line 50, numbers format ( according to the journal guidelines; where there are five or more digits to the left of the decimal point, use a comma to separate every three digits and significant figures). I also made symbols bigger in figure 3.

I will upload the corrected version based on reviewers remarks.  

Reviewer 2 Report

This paper reports the results of groundwater sampling and analysis in the Essaouira Basin, Morocco. In total, 113 samples were collected between 2015 and 2019 from 50 wells. Inorganic water quality parameters were analyzed either onsite or offsite depending on the parameters. Water quality change was shown by Piper diagram and Chadha diagram, indicating a shift from Ca-Mg-Cl type to Na-Cl type due to sea water intrusion.

TDS was used as an indicator for suitability of the groundwater to either drinking or irrigation uses.

Although the methods employed in this study are conventional, and no novelty in them, the results obtained in this research are useful for understanding the current situation of groundwater quality and future planning of groundwater management in the Essaouira Basin.

Fig 1. X-axis.

Please with one-year interval rather than 10 months interval.

  1. 49-50. The hyphens must be replaced with em-dash symbols.

  1. 224-226. Why the EC values are almost ten times different in the groundwater samples.

Please add explanation, especially with regard to the well sampling locations and underlying geochemical processes.

  1. 263-269. Fig 4.

Very little or no comment on Fig 4. Add more comments or discussion on Fig 4, or remove it if not needed.

Tables 2, 3, 4 and 5.

From these tables, it was not able to see the changes of water quality compliance to meet the drinking water or irrigation water standards. This might be due to short time period of the sampling (4 years). Thus, this should be added in to the text, and suggestion to a longer monitoring should be also added to the recommendation for future studies.

Author Response

Thank you for the interesting remarks.

Would you please find attached an attempt to reply to your remarks and comments.

I will upload the corrected version based on reviewers remarks. 

Reviewer 3 Report

The Authors describe in the manuscript characterization of groundwater quality in one of the coastal areas in Morocco (Essaouira basin). An extensive hydrogeochemical analyses were performed in combination with GIS tools (ArcGIS). The presented results are based on the data obtained during measurements campaigns in 2015-2019.

Composition of the article is correct, typical for a scientific paper (Introduction, Materials and Methods, Results and Discussion, Conclusion). The Authors have included an extensive reference list (57 items). The manuscript is written in clear, understandable English and is interesting to the reader. The presentation and interpretation of results is supplemented by numerous good quality figures.

The overall rating of the manuscript is good, the materials and methods are described sufficiently, the results, discussion and conclusion are correct.

There are no detailed remarks to the manuscript except two:

  • the Table 1 and some Figures failed when producing the pdf file, so they could not be seen fully and – consequently - evaluated.
  • the Authors should consider eventual movement of Appendix A from the main body of the manuscript to the additional materials because of its big size.

Author Response

Thank you for your remarks. 
I moved the appendix A to the supplementary materials and I will upload the corrected version based on reviewers remarks.

Round 2

Reviewer 1 Report

The manuscript was enhanced regarding its strucutre and contents. Some figures are still mosplaced. Can be accepted after this minor corrections.